# Input Digitization of the Manufacturing Industry and Carbon Emission Intensity Based on Testing the World and Developing Countries

**DOI:** 10.3390/ijerph191912855

**Published:** 2022-10-07

**Authors:** Hui Fang, Chunyu Jiang, Tufail Hussain, Xiaoye Zhang, Qixin Huo

**Affiliations:** School of International Trade and Economics, Shandong University of Finance and Economics, Jinan 250014, China

**Keywords:** digital economy, input digitization, carbon emission, manufacturing

## Abstract

Facing the increasingly deteriorating climate, carbon emission reduction has become a global consensus. In particular, as an industry with very serious pollution emissions, the manufacturing industry is under enormous pressure to reduce environmental consumption. At the same time, against the background of rapid digitization development, the production and organization of the manufacturing industry have greatly changed, which also provides new research ideas for global carbon emission reduction. Based on the panel data of 40 major economies in the world, this paper calculates the degree of input digitization of the manufacturing industry using the input–output method and constructs a triple fixed effect model to analyze the impact of manufacturing’s input digitization on its carbon emission intensity from the perspective of the world and developing countries. The research finds that, first, on the global level, input digitization significantly reduces the carbon emission intensity of manufacturing, and the effect of carbon reduction increases gradually over time, with a noticeable industry spillover effect. Second, the test results from developing countries show that the relationship between digital input from developed countries and manufacturing’s carbon intensity in developing countries presents an inverted U shape. Third, heterogeneity analysis shows that digital input has the most obvious effect on carbon reduction in the pollution-intensive manufacturing sector. Tracking the sources of digital input, it is found that digital input from high-tech economies has the most obvious effect on carbon reduction. The paper takes the lead in clarifying the impact of digitization on carbon emissions from the manufacturing sector, expands the existing research on the digital economy and the environment, and also makes a theoretical contribution to global carbon emission reduction.

## 1. Introduction

With the rapid evolution of the new scientific and technological revolution, the latest generation of information and communication technology has achieved an integrated breakthrough. Frontier Technology, including Big Data, Cloud Computing, and Artificial Intelligence, is promoting the reform of resource allocation and industrial upgrading with unprecedented breadth and depth [1]. The Global Digital Economy White Paper (2022) shows that the added value of the digital economy in 47 major countries around the world has reached USD 38.1 trillion, accounting for 45.0% of GDP, indicating the growing importance of digital elements [2]. In order to reshape the competitive advantage, the manufacturing industry has introduced a large number of digital assets as new production factors and applied digitization on the strategic level [3,4]. For example, the industrial digitalization scale of China has achieved USD 5.85 trillion, occupying 81.7% of the digital economy. The digital transformation of the manufacturing industry realizes renewed value creation and remodeling through the reorganization of resources and the innovation of production modes [5]. Meanwhile, it also exerts a new impact on resource utilization and the environment. At present, facing the increasingly deteriorating climate and environment, carbon emission reduction has become a global consensus. The EU plans to implement a carbon border tax in 2026 and proposes to take the lead in achieving “carbon neutrality” by 2050. The United States returned to the Paris Agreement and proposed to achieve the goal of a 100% clean energy economy and net-zero carbon emission by 2050. China has also responded positively to the international call, suggesting to strive for the peak of carbon dioxide emissions by 2030 and achieve carbon neutrality by 2060 [6]. As the pillar industry of the economy, the manufacturing industry has the characteristics of high input and high consumption, and it is an important source of greenhouse gas emissions. International Energy Agency (IEA) statistics show that, in 2019, the carbon emissions of the manufacturing industry reached 6254 million tons, accounting for 19% of global greenhouse gas emissions, surpassed in amount only by the energy industry. Manufacturing faces severe pressure on carbon reduction. In particular, the developing economies represented by China have undertaken major manufacturing activities in the world, making it more difficult to complete their energy conservation and emission reduction tasks. The IEA stated that, in 2019, the carbon emissions of developing economies accounted for 58% of the world. Therefore, the issue of carbon emission in the manufacturing industry should be given full attention, and it is essential to explore the path to reducing carbon emissions in manufacturing, especially in developing economies.

Against the background of the rapid development of the digital revolution, more and more countries are paying attention to the application of digital technology in environmental protection. For example, the 2021 China Double Carbon Strategy and Energy Digitization Forum put forward the view that digital technology will accelerate the process of the energy revolution and help the country achieve the goal of carbon peak and carbon neutralization, which affirmed the significant position of digital technology in carbon emission reduction strategy. However, as a new production factor, whether digital input can truly realize environmental sustainability through integration with the manufacturing industry is a question worthy of deep consideration. Generally speaking, digital input should positively affect the ecology and the environment through less resource consumption and the substitution of pollution elements [7]. However, will resource consumption caused by the manufacturing industry’s digital transformation aggravate the burden on the environment? From the perspective of developing countries, which have been undertaking the work of processing and manufacturing for the long term, could digital input reverse the current situation of high energy consumption and high pollution in the manufacturing industry? It is of great theoretical and practical significance to clarify these issues for achieving national environmental goals and accelerating the process of world carbon emission reduction.

The main structure of this paper is as follows: The second part reviews and combs the relevant literature, including carbon emission and digitization in manufacturing, as well as digital investment and FDI. The third part conducts the theoretical analysis and proposes research hypotheses. The fourth part constructs an econometric model, describes the calculation methods of key variables, and introduces the data sources. The fifth part provides the corresponding empirical test results, including benchmark regression results, robustness test results, mechanism test results, endogeneity test results, and extended analysis. The sixth part further extends the analysis for developing countries based on the hypothesis proposed in the third part. The seventh part summarizes the full text and puts forward corresponding policy suggestions and also discusses the shortcomings and future perspectives of this research.

## 2. Literature Review

With the continuous deterioration of the global climate and environment, the issue of carbon emissions has been widely concerning. As an industry with severe energy consumption and pollution, manufacturing is an important source of global greenhouse gas emissions. Existing literature has fully explored the field of carbon emission in the manufacturing industry, and its research is mainly divided into the following two categories: The first category focuses on the measurement of carbon emission in the manufacturing industry, which includes the Logarithmic Mean Divisia Index (LMDI) method [8], the Multi-Regional Input–Output Method (MRIO) [9], etc. The second type of literature mainly focuses on the analysis of the influencing factors of carbon emissions. Specifically, in the existing research, some scholars use the Index Decomposition Analysis (IDA) and Structural Decomposition Analysis (SDA) methods to investigate the actual contribution of various factors to carbon emissions in the manufacturing industry. For example, Li et al. took China’s manufacturing industries as research samples and decomposed the impact of various factors, including economic development, population size, and technical progress on carbon emission levels [6]. While other researchers tend to focus on the impact of a single factor on carbon emissions, such as scale expansion, structural adjustment, and investment changes [10,11], the main driving force behind them is still the change in energy or resource consumption. Therefore, improving production efficiency [12] and promoting technological progress [13] are effective ways to reduce carbon emissions.

Regarding research on the manufacturing industry’s digital transformation, the early literature mainly focused on the perspective of technology. Scholars used to believe that digital transformation refers to the improvement of production efficiency and enterprise performance through digital technology [1] or the improvement of the management and decision level by using information technology such as ERP [14]. In recent years, with the development of new digital technologies such as Big Data and Cloud Computing, the importance of digital technology in enterprise management has been further strengthened, and the connotation of industrial digitization has expanded further. For example, Ilvonen believes that industrial digitization refers to applying digital technology to products, production, and services, promoting the restructuring and transformation of enterprise production and operation mode to gain competitive advantages [15]. No matter how the definition of industrial digitization changes, the constant is that the existing literature attaches great importance to the role of information or data and regards it as a new production factor, believing that it has an impact on industrial production, organization, and management through its penetration and integration with traditional factors [5,16]. This has been established by policies in some countries, such as China, which officially established the status of data as a production factor in 2019. In this paper, we mainly refer to the research of Schallmo and Williams [17] to define digitalization in the manufacturing industry. They elaborate on digital transformation and break it down into two processes: digitization and digitalization, where digitization represents the process of converting analog information to digital information. In other words, it is the process of acquiring information and data through digital technologies such as information and communications technology (ICT). Digitalization is a fundamental change made to business models based on newly acquired knowledge gained via digitization initiatives [17]. Clarifying the distinction between these two definitions will be of great benefit to our research. Therefore, by combing the existing literature, we define digitalization in manufacturing as the transformation of business processes such as design, production, and warehousing based on newly formed knowledge obtained through digital methods, which collect data and store them as crucial factors, thus gaining comparative advantage and creating new value. This definition is not limited to early or new digital technologies, but it applies to the analysis in this paper as an inclusive concept, so what we investigate about digital technologies in this paper regards generic digital technologies, as represented by ICT.

As for the research on the impact of digitization on carbon emissions, the existing literature is mainly carried out at the national or regional level [7,18,19,20]. The relationship between digital technology and energy consumption is complex. The development of digital technology has both positive and negative effects on the environment [7,18]. Haseeb et al. investigated the impact of ICT on the environment based on the panel data of BRIC countries and found that ICT positively contributes to environmental quality through internet development and technological progress [21]. However, Shvakov and Petrova used the data from countries with high digital competitiveness and found that economic digitization will impose a more significant burden on the environment by improving economic expansion and energy consumption [20]. Furthermore, based on China’s provincial or urban panel data, some researchers have concluded that the development of the digital economy could significantly reduce urban carbon emissions by improving energy structures or promoting technological progress [22]. Some studies also suggest that the transformation cost brought on by digitization and the consumption of power resources may increase carbon emissions [23]. In light of the effects of two different directions, some scholars have also found that there is an inverted U-shaped relationship between carbon emissions and ICT [24]. However, due to restrictions during data acquisition on digitization, the existing studies on the environmental effects of manufacturing’s digitization are mainly qualitative analyses or professional technical reports [25,26], which lack empirical examination.

Compared with developed countries, developing countries’ imperfect infrastructure and a relatively late start in the digital economy have led to the immature development of digital technology, which results in the digital dependence on developed countries [27]. Furthermore, technology transfer accompanied by industry undertakings will also lead digital input to flow inward [28]. Therefore, a considerable part of developing countries’ digitalized input to the manufacturing industry comes from developed countries. Compared with the digital input from home, the digital input from developed economies has both high technical content and monopoly characteristics [27]. Meanwhile, the digital input inevitably contains part of the digital investment, which could significantly improve the technical level of manufacturing enterprises but may also lead to the problem of cost increase [29]. Especially when a digital investment comes from other economies, it shall also possess the attributes of foreign direct investment. Existing studies have usually regarded the impact of FDI on the environment as positive (Pollution Halo Hypothesis) or negative (Pollution Shelter Hypothesis) and generally consider a linear relationship between FDI and the carbon emissions of host countries and prove the above two hypotheses based on this [30,31]. However, when the investment has digital elements, the relationship between them may become more complicated. As an investment with highly knowledge-intensive characteristics, digital investment includes the integration of advanced manufacturing technology, intelligent technology, and information technology, and possesses a significant technology spillover effect [32]. Simultaneously, considering the cost brought on by digital transformation [33] and the productivity paradox of information technology [34], the Pollution Haven and Pollution Halo effect may both be reflected in digital investment. Therefore, there may be a complex nonlinear relationship between digital input from developed countries and carbon emissions in developing countries. To investigate the environmental effects of input digitization more comprehensively, this paper will further study the relationship between digital input from developed countries and the carbon emission intensity of developing countries.

Therefore, after combing the relevant literature, this paper finds that, first, studies on the impact of digitalization on carbon emissions are mainly conducted on the national or regional level, and few scholars focus on the impact of digitalization on carbon emissions in the manufacturing sector, while the way digital technology represented by ICT works on the manufacturing industry is quite different. Even if the research is performed, the scholars only analyze it qualitatively and lack empirical studies. Second, in the manufacturing sector, large quantities of papers have examined the economic effects and industrial upgrading effects of ICT, but there is a dearth of studies on the environmental effects of digitalization. Third, the existing literature pays less attention to the differences between the impact of different sources of input on production and the environment. The inputs are usually regarded as a whole while conducting the research. Therefore, based on the deficiencies in the above research, this paper takes carbon emission intensity as the research object and uses the empirical method to investigate the impact of input digitization on manufacturing’s carbon emissions from multiple perspectives. The marginal contribution of this paper is as follows: Firstly, by focusing on the lack of quantitative research on the environmental benefits of digitization, this paper empirically examines the carbon reduction effect of input digitization in manufacturing for the first time and proves it through rigorous intermediary mechanisms, which provides empirical support for theoretical research on the promotion of carbon emission reduction with digitization. Secondly, the paper not only tests the carbon emission reduction effect of input digitization on the global level but also verifies the relationship between digital input and the carbon emission intensity of host countries from the perspective of developing countries, which traces the impact of different sources of input and clarifies the path for carbon emission reduction from different angles. Thirdly, the paper conducts various heterogeneity analyses from multiple angles and examines the dynamic effects and spillover effect of digital inputs’ impact on carbon emission reduction to enhance the comprehensiveness and robustness of the research.

## 3. Theoretical Analysis and Research Hypothesis

Generally speaking, the input digitization of manufacturing has significantly positive effects on the environment through the following three mechanisms:

The first mechanism is to achieve carbon reduction by improving production efficiency. With the introduction of digital technology, data acquisition, storage, and transmission costs are significantly reduced [35]. Manufacturing enterprises can arrange production activities more reasonably based on data and information. According to the research of Bartel et al., the use of ICT in the manufacturing industry will improve the efficiency of all stages of the production process by reducing setup times, run times, and inspection times. Therefore, digital technology will realize the reorganization of business processes, and significantly improve production and operation processes, hence reducing management costs and improving production efficiency [36], which is consistent with the logic of Process Reengineering Theory. After combining traditional production factors such as capital and labor with data, this new element may achieve several times greater efficiency than the previous production capacity [37], thus reducing the capital and resource consumption under the same production quantity and leading to a reduction in the resource consumption of per unit output. Previous studies have also proved that the improvement of TFP has a significant inhibitory effect on carbon emissions because of efficient resource utilization [12,38]. Specifically, the increased productivity and rationalization of production organization resulting from digital inputs help to direct the efficient allocation of resources, which leads to increased energy efficiency and lower carbon emissions [39]. In addition, digital technologies allow manufacturing enterprises to track consumers’ personalized needs and build exclusive databases and then carry out personalized product customization and marketing from the consumer side [40]. As mentioned above, digital technologies have greatly reduced setup costs, while setup costs are a bigger fraction of unit costs for customized products, so the reduction in unit costs due to a reduction in setup time is greater for customized products than for commodity products [36]. This helps manufacturing enterprises shift from the large-scale standardized production of homogeneous products to a customized production mode that meets the heterogeneous needs of consumers, improving the added value as well as production efficiency [40]. Furthermore, the transformation of the production mode also greatly reduces the resource waste caused by invalid inventory and benefits the environment. This means that the transformation of the production mode brought about by digital input is different from the simple scale expansion caused by traditional technological innovation, but it produces the environmentally friendly innovation of production and organization. Moreover, according to Technical Innovation Theory, innovation is the re-combination of production factors, which means introducing a new combination of production conditions and production factors that did not exist before into the production system. The introduction of ICT makes data a completely new production factor. Massive amounts of data are gathered in the process of economic development, and they help traditional production factors to achieve product innovation and production method breakthroughs via the fast feedback of market information and process reorganization [41]. Furthermore, the data’s characteristics of easy storage and copying allow them to break through the limitations of existing production factors and provide support for enterprises to achieve product innovation [4]. Innovation in products and production leads to the improvement of production efficiency, which in turn results in efficient use of resources and carbon neutralization.

The second mechanism is to achieve carbon reduction by improving energy efficiency. Firstly, the introduction of digital technology helps collect massive amounts of market information. The fast feedback of demand information can help enterprises realize the constant adjustment of production structures to manage energy consumption accurately and improve energy efficiency [42]. Secondly, what needs to be mentioned is that energy has high substitution flexibility, and the possibility lies in the substitution between energy and other production factors. In particular, capital has a considerable substitution space for energy, especially physical capital, showing a stronger substitutive relationship. Furthermore, with the increase in neutral technological progress parameters, the substitution becomes stronger [43]. This indicates that physical capital, such as manufacturing machines with higher technology content, will achieve greater substitution for energy. The introduction of digital elements has improved manufacturing production technology and derived digital manufacturing equipment such as numerical control machines, which has greatly reduced carbon emissions by improving production efficiency and replacing a large number of polluting elements such as energy. In addition, as a new production factor, data have the characteristics of cleanness, low cost, and easy sharing compared with traditional resources, which helps to reduce pollution emissions further and change the energy consumption structure in the process of element substitution [44]. Specifically, ICT offers various functions such as the substitution of virtual processes for physical processes, system monitoring with censoring tools, data transmission and processing, and efficient equipment control. ICT enhances the decoupling of economic activities from energy use through these functions [45]. Finally, ICT enables manufacturing enterprises to access the internet and the native area network in the production process and optimize energy parameters through the local industrial energy network. Meanwhile, as mentioned above, the rationalization of the production process also brings about the optimization of the energy use structure [46]. Martynenko believes that digitalization delivers the ecological modernization of production, which can save resources and secure industry and societal sustainability [47].

The third mechanism is to achieve carbon reduction by improving information transmission. Negroponte believes that digital technology reduces the cost of information storage, replication, and transmission and improves the traceability of content in innovation activities [35]. Other studies also prove that digital technology has dramatically reduced the cost of information transmission and improved the speed and scope of information dissemination [48]. According to Spillover Theory, the spread of information will produce extensive externalities and spillover effects. For example, while testing the role of knowledge dissemination, Wiel et al. introduced technology spillover variables into the production function and found that information technology spillover can significantly and continuously promote the improvement of productivity [49]. Furthermore, Absorptive Capacity (ACAP) Theory indicates that information technology can enhance firm innovation by facilitating the creation of patent inventions and the introduction of new products and services into the market. ICT can help create new knowledge by merging, categorizing, reclassifying, and synthesizing existing knowledge [45]. Existing studies have shown that there is a significant substitution relationship between technological progress and carbon emissions [50,51]. Technological innovation brought about by digital elements can improve carbon emission efficiency by improving production efficiency and environmentally friendly technology [52]. Inevitably, technological progress may also lead to the rebound effect, which creates new demands for energy due to industry expansion and partially counteracts energy efficiency improvement [53,54]. However, as mentioned above, the innovation in production patterns brought about by digital technology is different from the disordered expansion led by traditional technological advances. Furthermore, digital input tends to promote resource-saving, biased technological progress, and the development of environmental protection technology [55], which indicates that the rebound effect of digital technology shall be relatively weak. Moreover, information technology can promote the transformation and upgrading of traditional industries by reducing material dependence and reshaping the production structure, which greatly promotes carbon reduction [56].

With the internet and sharing networks, enterprises’ latest technologies for environmental protection or energy innovation will spread more quickly in the industry through the digital channel. Positive environmentally friendly technology spillover is conducive to reducing the industry’s resource consumption and carbon intensity [57]. The information transparency caused by the development of digital technology will also make enterprises face greater competition and environmental protection pressure, which will push them to upgrade production and environmentally friendly technology by increasing innovation investment. Moreover, the utilization of ICT enables energy-using information to be shared within industries or regions, realizing the energy supply segment’s intensification, digitization, and refinement. This helps alleviate the information asymmetry between the energy supply side and the demand side and makes the energy supply process more reasonable, thus avoiding energy waste and overproduction and improving the efficiency of resource allocation and carbon emission [48]. Therefore, the improvement of information transmission helps the industry enhance innovation and production efficiency, replace outdated polluting technology, and promote the development of environmental protection techniques, leading to a reduction in resource consumption and pollution emissions [58].

Based on the analysis above, this paper puts forward the following hypothesis:

**Hypothesis** **1** **(H1):**
*Input digitization can reduce the carbon emission intensity of the manufacturing industry. Specifically, it is mainly achieved by improving production efficiency, energy efficiency, and information transmission.*


Although input digitization has many positive effects on the environment, there may be differences in how digital input affects the environment in developing countries. Since the 1960s, developing countries have attracted and undertaken a large number of processing and manufacturing segments with high energy consumption and pollution from developed countries due to abundant energy, labor, land, and other resources, as well as relatively weak environmental regulations. The advanced production technology and management skills of multinational companies will also transfer in the process of the industrial undertaking [28]. Due to the significant role of digital technology in reducing costs, expanding the market, and improving production efficiency [59], digital technology and investment have become an essential part of international technology transfer, such as investment in computers, communication equipment, and the internet, as well as the provision of telecommunication and data processing services. Therefore, digital input from developed countries is likely to be a necessary investment or technology transfer provided by multinationals to carry out production activities more effectively. These production activities may belong to labor-intensive or resource-intensive manufacturing industries with high energy consumption. Therefore, this part of the digital input may not be conducive to environmental improvement in developing countries and may even cause further environmental pollution.

Furthermore, most countries have taken digital protection measures to protect domestic data flow and restrict data and core technology transfers to other countries. Digital barriers and upstream technological monopolies hinder foreign digital input’s knowledge spillover and technology upgrading effect, limiting its contribution to reducing carbon emissions [60]. Furthermore, excessive industrial transfer accompanied by digital investment may even lead developing countries into a low-end lock-in and being captured in processing and assembling industries with low value-added [61], further inhibiting the positive environmental effects brought on by digitization.

On the other hand, in consideration of developing countries’ own disadvantages, weak environmental regulations and limited knowledge absorption capacity will also restrict the environmental effects of digital input. The digital input provided by developed countries will improve the production efficiency and economic effect of the manufacturing industry in developing countries, resulting in a rapid expansion of industrial scale, and weak environmental regulations cannot effectively restrain the environmental pollution caused by the expansion of production activities, eventually leading to an increase in resource consumption [62]. Moreover, the low level of human capital in developing countries leads to their weak ability to absorb advanced digital technologies, while not allowing a rapid shift in production patterns and leading to the rebound effect of technical progress. Therefore, the effects of technology improvement and carbon emission reduction are both restricted in the short term [63]. In addition, local enterprises’ absorption of and research on transferred digital technologies will generate a large demand for talent, funds, and other innovative elements, which will lead to an increase in costs and, consequently, will not be conducive to environmental improvement in the short term [64]. Therefore, this paper puts forward the following hypothesis:

**Hypothesis** **2** **(H2):**
*In the short term, digital input from developed countries has an inhibitory effect on reducing the carbon emission intensity of the manufacturing industry in developing countries.*


Although digital input from developed countries may come with the motivation of carrying out manufacturing activities effectively, even the minimum technology transfer also provides a technical basis for the manufacturing industry in developing countries. Technology spillovers can be realized through the demonstration and imitation effect, personnel flow effect, and connection effect [65] to improve environmental quality. Moreover, access to digital technology and internet applications will expose enterprises to foreign patents and advanced technologies, thus causing technology spillover and the driving effect to emerge in local enterprises [66]. In addition, the improvement of production efficiency brought on by digital input and the companying industrial transfer helps developing countries to embed in the global value chain, which will have a positive scale effect, spillover effect, and competition effect on the domestic manufacturing industry [61]. It will promote independent R&D innovation and the industrial upgrade of developing countries, consequently having a beneficial influence on the environment in the long term. Simultaneously, the introduction of digital technology may be the beginning of local digital industry, driving the emergence and development of industries such as computer manufacturing, telecommunications services, and other emerging industries, consequently improving the digital level of the domestic manufacturing industry and producing positive environmental effects. Moreover, in the long run, when the R&D investment of enterprises reaches a certain level, digital technology will lead to the rapid improvement of innovation efficiency. As intellectual assets, digital technologies such as digital software and information management systems have meager marginal costs [67]. After being absorbed and mastered by local enterprises, they can be used cheaply by branches or put into technology transactions with other enterprises. They will produce a significant technology spillover effect and promote technological progress and digital transformation, thus realizing the elimination of pollution elements. Based on the above analysis, this paper puts forward the following hypothesis:

**Hypothesis** **3** **(H3):**
*With the passage of time, the effect of developed countries’ digital inputs on reducing carbon emissions in developing countries’ manufacturing industries will continue to increase and eventually exceed its environmental inhibitory effect, which makes the digital input from developed countries and the carbon emission intensity of developing countries an inverted U-shaped, nonlinear relationship.*


## 4. Empirical Models and Data Processing

### 4.1. Empirical Model

To articulate the environmental effect of digitization and verify the hypothesis above, this paper constructs the following fixed effect model:(1)$$Carbon_{cit}=\beta_{0}+\beta_{1}DIG_{cit}+\beta_{2}Controls_{cit}+uc+ui+ut+\varepsilon cit$$

In Equation (1), the subscripts *c*, *i*, and *t* represent the economy, industry, and year respectively; $Carbon_{cit}$ is the carbon emission intensity of the manufacturing industry; $DIG_{cit}$ represents the degree of input digitization of the manufacturing industry; and *Controls* means control variables, which include: (1) Value added (Va), to control the impact of manufacturing scale on carbon emissions. (2) The proportion of exported domestic value-added (Dvafs), expressed by the ratio of exported domestic value-added from manufacturing to the industry’s total output. We use this index to examine the impact of foreign trade. (3) Capital stock per labor (*CL*), expressed by the ratio of the real fixed capital stock and the number of employees in the manufacturing industry. The capital stock represents the equipment perfectness and modernization degree of the enterprise. (4) Energy intensity (Energy), expressed by the ratio of energy consumption to the total output of the manufacturing industry. (5) Energy consumption construct (Construct), expressed by the ratio of non-clean energy consumption to total energy consumption. (6) Foreign direct investment (FDI). The impact of FDI on carbon emissions has been widely recognized. Since FDI data at the industry level cannot be obtained, it is quantified by the proportion of actual FDI in the GDP of each economy. (7) Environmental regulation intensity (Env). This paper selects the energy consumption per unit GDP of each economy to represent the intensity of environmental regulation from the output perspective. The larger the value, the smaller the intensity of environmental regulation. uc, ui, and ut represent country, industry, and time-fixed effects. εcit is a random error term.

### 4.2. Variable Definition

#### 4.2.1. Dependent Variable—Complete Carbon Emission Intensity

Referring to the work of Pan and Wei [9] and Huang and Xie [68], this paper uses the MRIO method to calculate the carbon emissions of the unit final output of the manufacturing industry, that is, the complete carbon emission intensity.

According to the world input–output model, the paper supposes that there are *G* economies in the world, and each economy has *N* industrial sectors.
(2)$$X= [\begin{array}{c} X^{1}\\ X^{2}\\ \vdots \\ X^{G} \end{array}]= [\begin{array}{cccc} A^{11} & A^{12} & \cdots & A^{1G}\\ A^{21} & A^{22} & \cdots & A^{2G}\\ \vdots & \vdots & \ddots & \vdots \\ A^{G1} & A^{G2} & \cdots & A^{GG} \end{array}] [\begin{array}{c} X^{1}\\ X^{2}\\ \vdots \\ X^{G} \end{array}]+ [\begin{array}{c} Y^{1}\\ Y^{2}\\ \vdots \\ Y^{G} \end{array}]$$

The above equation can be simplified as X=AX+Y. Where *X* is the total output column vector of each economy, *Y* is the final product column vector of each economy, and *A* is each economy’s direct consumption coefficient matrix. $A^{sr}$ represents the direct consumption coefficient matrix of economy *r* to economy *s*.

On this basis, the carbon emission coefficient of industry *i* in economy *s* is set as:(3)$${d_{i}}^{s}=\frac{{{Car}_{i}}^{s}}{{X_{i}}^{s}}$$

In Equation (3), ${{Car}_{i}}^{s}$ is the carbon dioxide emissions of industry *i in* economy *s*, and ${X_{i}}^{s}$ is the total output of industry *i in* economy *s*, so we can define the row vector of the carbon emission coefficient of each economy in each industry as follows:(4)$$D= [d_{1}{}^{1},d_{2}{}^{1},\cdots ,d_{I}{}^{1};d_{1}{}^{2},d_{2}{}^{2},\cdots ,d_{I}{}^{2};\cdots ;d_{1}{}^{G},d_{2}{}^{G},\cdots ,d_{I}{}^{G}]= [d^{1},d^{2}\cdots ,d^{G}]$$

Based on this, we can obtain the matrix of each economy’s complete carbon emissions with respect to the final output:(5)$$T=\hat{D}B\hat{Y}= [\begin{array}{cccc} d^{1}B^{11}Y^{1} & d^{1}B^{12}Y^{2} & \cdots & d^{1}B^{1G}Y^{G}\\ d^{2}B^{21}Y^{1} & d^{2}B^{22}Y^{2} & \cdots & d^{2}B^{2G}Y^{G}\\ \vdots & \vdots & \ddots & \vdots \\ d^{G}B^{G1}Y^{1} & d^{G}B^{G2}Y^{2} & \cdots & d^{G}B^{GG}Y^{G} \end{array}]$$

In Equation (5), $\hat{D}$ and $\hat{Y}$ represent the diagonalization matrix of carbon emission coefficient vector *D* and final product vector *Y*, respectively. *B* is the Leontief inverse matrix and $B=(I-A{)}^{-1}$, where *I* is the identity matrix. By summing the matrix elements by column, we can obtain the complete carbon emissions of the final output of each economy. Further, by substituting the $\hat{Y}$ for the unit matrix, *I*, the complete carbon emission intensity of each economy can be obtained:(6)$$Carbon=DBI=DB= [\sum_{m=1}^{G}d^{m}B^{m1}\;\sum_{m=1}^{G}d^{m}B^{m2}\;\cdots \;\sum_{m=1}^{G}d^{m}B^{mG}]$$
where $\sum_{m=1}^{G}d^{m}B^{mn}$ is a N-order row vector containing *N* industry sectors. Now, we can obtain the complete carbon emission intensity of 14 manufacturing industries from 40 economies. This indicator not only reflects the carbon emissions caused by the direct consumption of the final product but also includes the carbon emissions caused by the intermediate input consumed in the production of the final product.

#### 4.2.2. Independent Variable—The Degree of Input Digitization

The data in this paper are mainly from the WIOD 2013 database, which uses ISIC Rev3.1 as the basis for industrial classification. Therefore, this paper also uses this classification criterion and refers to the work of Zhang and Yu [69] and Xu and Zhang [70] to classify digital industries according to the classification of information and communication technology departments in ISIC Rev3.1. Considering the existence of digital and non-digital sectors in specific industries, we use the data released by the United Nations Conference on Trade and Development (UNCTAD) and the UN Comtrade database to split the digital department. The specific digital industries are shown in Table 1.

According to the coefficient shown in the table, the digital sector can be separated from the corresponding industry. Since the input–output data of industry 72—Computer and related industries are not included in WIOD 2013 and this industry occupies an important position in the measurement of input digitization, this paper uses the input–output data in the WIOD 2016 database to merge the industry according to the ISIC rev3.1 standard and then brings 72—Computer and related activities into the input–output data of WIOD 2013. After that, we finally obtain the intermediate input matrix of 40 economies and 42 sectors and can calculate the input digitization level of the manufacturing industry based on this.

Referring to the method of Liu et al. [71], this paper uses the input–output method to measure the degree of input digitization in manufacturing, which includes the direct and indirect consumption coefficient methods. The direct consumption coefficient refers to the input from the digital sector consumed by the unit output, that is, the direct consumption of digital input by the manufacturing sector. The complete consumption coefficient reflects the direct relationship between departments and includes the indirect economic connection. The calculation method is as follows:(7)$$DIGdj=adj+\sum_{m=1}^{N}admamj+\sum_{l=1}^{N}\sum_{m=1}^{N}a_{dl}a_{lm}a_{mj}+\ldots$$

The left side of Equation (7) represents the complete consumption coefficient of department *j* to digital department *d*. The first item on the right side of the equation is the direct consumption of department *j* to department *d*. The second item is the first indirect consumption of department *j* to department *d* through department *m*, the third item is the second indirect consumption of department *j* to department *d* through departments *m* and *l*. The rest can be deduced by analogy. By summing up the complete consumption coefficient of the manufacturing sector to the abovementioned digital sectors, we can obtain the degree of input digitization of manufacturing industry *j*, $\sum_{d=1}^{n}DIG_{dj}=DIG_{j}^{G}$, in which *n* is the number of digital departments.

### 4.3. Stylized Facts Analysis of Input Digitization in Manufacturing

According to the measuring method of input digitization described above, this paper calculates the degree of input digitization of the world’s manufacturing industry from 2005 to 2011 and analyzes its stylized facts and development trend.

Figure 1 describes manufacturing’s input digitization level in different economies of the world. Overall, the world’s degree of input digitization is rising steadily, and the level of input digitization in developing economies is lower than in developed economies. Focusing on China, its manufacturing industry’s degree of input digitization was lower than the average level of developing economies in the early stage, but the growth was extremely rapid. By 2011, it surpassed the developing economies and reached the world’s average level.

Figure 2 shows the input digitization level of different types of manufacturing industries in the world. We are informed that the degree of input digitization of each manufacturing industry shows an upward trend, and the degree of input digitization is higher in capital-intensive industries, followed by labor-intensive industries, and resource-intensive industries have a lower degree of digitization.

Figure 3 depicts different sources of digital input in developing economies. In the early stage, nearly one-third of the total digital input of developing economies came from developed economies, but as time went by, the digital dependence of developing economies on developed economies declined, and more digital input in the manufacturing industry comes from domestic and other developing economies.

### 4.4. Data Source Description

The data in this paper mainly came from the WIOD database and UNCTAD. Currently, only the WIOD 2013 database covers the carbon emission data on country–industry level. Furthermore, these data are indispensable for calculating the carbon emission intensity of the manufacturing industry. However, the sample period covered by this database is only up to 2011. Meanwhile, the earliest data related to the digital economy published by UNCTAD are for 2005 because the digital economy is a relatively new form of economy. Therefore, this paper selects 2005–2011 as the sample period. By studying the impact of ICT and other digital technologies on carbon emissions in manufacturing, we can draw general patterns of digital technology’s impact on the environment. Admittedly, the new generation of information technology, such as Big Data and Cloud Computing, is developing rapidly and is gradually being used in the manufacturing industry, but the theoretical analysis and the conclusions drawn from the research based on this data are equally applicable to the latest digital technology. As mentioned earlier, new digital technologies have, in essence, merely broadened access to information; in other words, they have improved the stage of “digitization” [17,72], but the nature of the way they work in the manufacturing industry remains fundamentally unchanged. Input digitization in the manufacturing industry is essentially the introduction of information as a production factor into the production and management process [5], and this paper has conducted theoretical and empirical analyses based on this concept. No matter how the data are collected, with the introduction of this new element, the manufacturing industry can realize digital design, digital manufacturing, and digital management, promoting the deconstruction of industries and value chains, the transformation of production and organization models, and the reconstruction of business models. Therefore, the essence of new digital technologies and their applications in manufacturing, also known as the stage of “digitalization”, has not changed fundamentally, so our theoretical analysis and empirical research are still reasonable.

In addition, the study of the environmental impacts of ICT and other digital technologies will also shed light on how the new generation of information technologies can be used more efficiently in the manufacturing industry, thus maximizing the environmental effects. Specifically, our theoretical analysis suggests that digital technologies will contribute to carbon reduction through three channels, improving production efficiency, energy efficiency, and information transformation, while new digital technologies will have a more prominent impact on the environment due to their stronger information-gathering functions and extensive penetration capabilities. By clarifying these transmission pathways, we will have a clearer understanding of how new digital technologies work on the environment and, thus, be more efficient in guiding them toward carbon reduction. On the other hand, digital technologies often lead to a rapid expansion of production scale and high R&D investment costs, resulting in increased energy consumption, and these problems are particularly serious for new digital technologies. Therefore, by studying the impact of digital technologies on carbon emissions and clarifying the possible negative effects in advance, effective means can be taken to avoid the negative environmental effects of new information technologies.

Since the data in the environmental account of the WIOD 2013 database is up to date for 2009, to make full use of the input–output data in WIOD, this paper refers to the work of Pan and Wei [9] and uses the trend method to complete the carbon emission data from 2010 to 2011, considering that the carbon emission of the manufacturing industry in each economy shows an obvious downward trend. The input–output data of 2010 and 2011 are still selected to calculate the carbon emission intensity of the manufacturing industry. The degree of manufacturing’s input digitization is calculated based on the input–output data in the WIOD database and the data published by UNCTAD. The complete carbon emission intensity of the manufacturing industry is calculated based on the input–output data in the WIOD database and carbon dioxide emission data in the environmental account. The data of value-added, energy intensity, capital stock per labor, and energy consumption construct in the control variables came from the environmental accounts and social accounts in the WIOD database. The nominal variables are deflated according to the price index to calculate the real value. The proportion of exported domestic value-added is from the UIBE-GVC database. Foreign direct investment data came from UNCTAD, and environmental regulation data came from the WDI database. This paper takes the natural logarithms of some variables to eliminate heteroscedasticity and winsorizes all variables in the 1% and 99% quantiles considering the influence of possible extreme values. Considering the possible multicollinearity among the variables, we test using the Variance Inflation Factor (VIF) method and find the maximum value of VIF is 3.25, which is far less than 10. Therefore, there is no multicollinearity problem in the model. Table 2 shows the classification of economies and manufacturing industries.

## 5. Empirical Results

### 5.1. Benchmark Regression Results

Table 3 shows the regression results of carbon emission intensity for the input digitization of manufacturing after adding control variables. It can be found that the level of input digitization, *DIG*, is always significantly negative, and in the case where all control variables are added, as shown in column (8), the carbon emission intensity of the manufacturing industry will decrease by 0.283% for every 1% increase in input digitization degree. Therefore, hypothesis 1 is proved: The increase in input digitization will significantly reduce the carbon emission intensity of the manufacturing industry. As pointed out by Bartel et al., digital technology will improve traditional industries’ production and operation processes, and plays a significant role in energy conservation and emission reduction [36]. The signs of control variables are consistent with expectations. The value-added of the manufacturing industry is significantly negatively correlated with carbon emission intensity, which means that the expansion of industry scale helps to improve the effect of carbon reduction, and indicates that the upgrading of industrial structure brought on by industry expansion plays a major role. The coefficient of exported domestic value-added is significantly negative, indicating that foreign trade’s structural upgrading and technology spillover effects are greater than the negative scale effect. The coefficient of capital stock per labor is significantly negative, indicating that capital investment is conducive to energy conservation and emission reduction. The correlation coefficient of energy intensity and energy consumption construct is positive, which makes logical sense. The negative coefficient of FDI and environmental regulation intensity indicates that the technology spillover effect of FDI plays a major role and that the weakening of environmental regulation will increase the carbon emission intensity in the manufacturing industry.

### 5.2. Robustness Test

#### 5.2.1. Replacing the Measurement Index of Input Digitization

To verify the reliability of the regression results, we adopt the relative indicator, and the core independent variable is replaced by the degree of reliance on input digitization of manufacturing. The calculation equation is as follows:(8)$${{DR}_{j}}^{G}=\sum_{j}(\sum_{d=1}^{n}DIG_{dj}/\sum_{k=1}^{N}complete_{kj})$$
where on the left side of the equation, $DR_{j}{}^{G}$ represents the reliance on input digitization of manufacturing *j* in economy *G*. On the right side, $\sum_{d=1}^{n}DIG_{dj}$ means the summing of the complete consumption coefficient of manufacturing sector *j* to digital sector *d* or the degree of input digitization of manufacturing *j*. $\sum_{k=1}^{N}complete_{kj}$ means the summing of the complete consumption coefficient of manufacturing sector *j* from all industries, *k*, where *N* represents the total number of industries. The index represents the relative importance of digital input in the total input. The regression result is shown in column (1) of Table 4, and the coefficient of complete carbon emission intensity is still significantly negative, without the sign changing.

#### 5.2.2. Replacing the Measurement Index of Complete Carbon Emission Intensity

In this part, we replaced the complete carbon emission intensity with the direct carbon emission intensity of the manufacturing industry, that is, the carbon emission per unit value-added, calculated as the ratio of carbon dioxide emission of the manufacturing industry to the value-added of the sector. Column (2) in Table 4 shows that the coefficient of input digitization is still significantly negative, but compared with Table 3, we can see that the impact of input digitization on the complete carbon emission intensity is more significant than on the direct carbon emission intensity. Because the direct carbon emission intensity only calculates the carbon emission caused by the increase in the unit output of the manufacturing industry, the complete carbon emission intensity takes into account the carbon emissions of other industries caused by the increase in manufacturing output. In other words, the complete carbon emission intensity also includes the carbon emission caused by the intermediate input consumed in the production of the final product, and the carbon emission intensity of these intermediate industries can also be reduced through digital input, thus reducing the carbon emission intensity of manufacturing industry further. Therefore, direct carbon emission intensity underestimates the digital input’s carbon reduction effect.

#### 5.2.3. Endogenetic Treatment

What needs to be considered is that the digital transformation of enterprises may come from the environmental pressure of the government and the public because of the increase in their carbon emissions. Therefore, there may be a reciprocal cause–effect relationship between input digitization and carbon emission intensity. Referring to the work of Huang and Xie [68] and Huang et al. [73], this paper selects the first-order lag term of manufacturing’s input digitization degree and the number of fixed telephones in 1980 as the instrumental variables to carry out 2SLS regression. Among them, the choice of the first-order lag term of input digitization as an instrumental variable is in common use and shall not be elaborated on in this paper. Since the number of fixed telephones is chosen because the digital technology represented by the internet mainly relied on the fixed telephone network in the early stage, the regions with a high penetration of fixed telephones, historically, are also likely to be the regions where the digital transformation starts first and also has a higher degree of development. Therefore, the selection of the number of fixed telephones as an instrumental variable can meet the correlation requirement. Furthermore, the number of fixed telephones is gradually decreasing due to the rapid development of the internet and digital technology, and it is hard for fixed telephones to influence the energy intensity and carbon emission efficiency of a region now. Columns (3) and (4) of Table 4 report the results of the 2SLS regressions with the first-order lag term of input digitization and the number of fixed phones as instrumental variables, respectively. First, in the first stage IV estimation, the regression coefficients of the instrumental variable on input digitization are 0.402 (*p* = 0.000 < 0.01) and 0.010 (*p* = 0.000 < 0.01), respectively, indicating a significant positive relationship between IV and the independent variables, which meets the correlation requirement. Second, as shown in columns (3) and (4) of Table 4, the regression results passed the unidentifiable and weak instrumental variable tests, indicating that the instrumental variables are valid, and the coefficients of input digitization are both significantly negative, proving that the benchmark regression results are still robust after considering the possible endogeneity.

#### 5.2.4. Sample Period Adjustment

Due to the limitation of data acquisition, while calculating the carbon emission intensity data of the manufacturing industry in 2010 and 2011, the data on carbon emission are calculated using the trend method. In order to ensure the robustness of the test results, the test is conducted again after eliminating the data of these two years. The test results are shown in column (5) of Table 4; we are informed that the sign and significance of key variables have not changed. The test results show that the regression is robust.

### 5.3. Mechanism Test

Based on the analysis above, this paper constructs the following test model:(9)$$Carbon_{cit}=\alpha +\beta_{1}DIG_{cit}+\gamma Controls_{cit}+u_{c}+u_{i}+u_{t}+\varepsilon_{cit}$$
(10)$$M_{cit}=\varphi +\beta_{2}DIG_{cit}+\gamma Controls_{cit}+u_{c}+u_{i}+u_{t}+\varepsilon_{cit}$$
(11)$$Carbon_{cit}=\delta +\beta_{3}DIG_{cit}+\beta_{4}M_{cit}+\gamma Controls_{cit}+u_{c}+u_{i}+u_{t}+\varepsilon_{cit}$$
where $M_{cit}$ is the intermediary variable, and other variables have the same meaning as the benchmark regression model in Equation (1). According to the mechanism analysis, the mediator variables include labor productivity (OL), energy intensity (Energy), and the level of information interaction (Inf). As a proxy of production efficiency, labor productivity is calculated as the ratio of the actual total output of the manufacturing industry to the number of employees. The data come from the social accounts in the WIOD 2013 database and have been deflated by the price index. The data of energy intensity are obtained from the control variables directly. The level of information interaction (Inf) is defined as the proportion of enterprises that interact through the internet. These data come from the weighted calculation of the B12 project in “ICT use in business by industrial classification of economic activity (ISIC Rev. 3.1)” under the digital economy catalog from UNCTAD.

Table 5 reports the results of the mechanism test. Column (1) corresponds to Equation (9), which is the same as the benchmark regression result of column (8) in Table 3. Column (2), column (4), and column (6) correspond to Equation (10), while column (3), column (5), and column (7) correspond to Equation (11). Firstly, columns (2) and (3) show the test results of the path of improving production efficiency. From column (2), we can see that the input digitization of manufacturing significantly improves labor productivity, and in column (3), when the carbon emission intensity is taken as the explanatory variable, the coefficient of input digitization and labor productivity are both negative, thus verifying the first mechanism. It is proved that input digitization can reduce carbon emission intensity by improving the production efficiency of enterprises. Secondly, columns (4) and (5) are the regression results while taking improving energy efficiency as an intermediary mechanism. We are informed that input digitization significantly reduces energy intensity, and energy intensity is positively correlated with carbon emission intensity, which illustrates that digitization reduces energy consumption per unit output, thus reducing the carbon emission intensity. Therefore, the second mechanism is proved. In other words, manufacturing digitization can reduce carbon emission intensity by improving energy efficiency. Finally, columns (6) and (7) show the test results of the third mechanism test. From column (6), we can see that the input digitization of the manufacturing industry greatly improves the level of enterprises’ information transmission. Meanwhile, in column (7), the coefficient of input digitization and the level of information interaction is significantly negative, which further verifies the last mechanism. That is to say, input digitization could reduce carbon emissions by improving information transmission. In order to make sure the robustness of the mediating effect test, we further perform the Sobel and Bootstrap tests on the mechanism test. As shown in Table 5, the Sobel test statistics for the three mechanisms of production efficiency, energy use efficiency, and information transformation improvement are −11.67, −2.847, and −14.53, respectively, whose absolute values are well above the critical value of 1.96. All of them passed the significance test at the 1% level. Meanwhile, the 95% confidence interval of the Bootstrap test does not include zero in any of the three mediation channels, which satisfies the condition of significant mediation effect.

### 5.4. Heterogeneity Analysis

#### 5.4.1. Heterogeneity Analysis Based on Different Types of Manufacturing Industry

To investigate the possible heterogeneous impact, this paper further divides manufacturing industries into labor-intensive, resource-intensive, and capital-intensive industries and examines the impact of input digitization on the carbon emissions of different manufacturing industries. Column (1), column (2), and column (3) in Table 6 show that the carbon reduction effect of input digitization on the resource-intensive industries is most obvious, followed by the labor-intensive industries, while the environmental effect of capital-intensive industries is not significant. The reason may be the high pollution characteristics of resource-intensive industries. Thus, embedding digital elements will significantly improve energy efficiency and reduce resource consumption. While labor-intensive industries have relatively fewer pollution emissions, the substitution of digital elements for pollution factors is relatively weak, and carbon reduction is mainly achieved through improving production efficiency. Regarding capital-intensive industries, the relatively advanced technology has limited the space for digital input to play a greater role in carbon reduction. At the same time, the limited pollution emission also leads to the weak substitution effect for pollution factors.

#### 5.4.2. Heterogeneity Analysis Based on the Input from Economies with Different Digital Levels

Due to different digital levels of economies, the technology spillover effect of digital input will be different, and there will also be differences in the effect on carbon reduction. Therefore, based on the Frontier Technology Readiness Index released by UNCTAD, this paper sorts the digital level of 40 economies and divides them into high-digital-level economies, medium-digital-level economies, and low-digital-level economies. The digital inputs from the three groups of economies are recorded as ${\mathrm{DIG}}_{H}$, ${\mathrm{DIG}}_{M}$, and ${\mathrm{DIG}}_{L}$. This part of the paper will test their correlation with carbon emission intensity. In Table 6, column (4), column (5), and column (6), respectively, show the impact of digital input from high-digital-level economies, medium-digital-level economies, and low-digital-level economies on carbon emissions in the manufacturing industry. We are informed that with the reduction in the digital technology readiness index, the effect of carbon emission reduction on digital input is also declining. This may be because digital input from high-tech economies includes more investment in digital infrastructures such as computers, internet construction, and intelligent manufacturing. This kind of input has a significant technology spillover effect, which is especially beneficial for the improvement of efficiency and carbon emission reduction [69]. The investment in advanced digital technology may be relatively small among the digital inputs from the other two groups of economies. In contrast, the input in digital media and transactions is relatively large, and the technology spillover effect is relatively weak. Hence, its impact on the carbon emissions of the manufacturing industry is relatively small.

### 5.5. Extended Analysis

#### 5.5.1. The Dynamic Analysis

Considering that there may be hysteresis in the impact of input digitization on manufacturing’s carbon emissions, this paper adds different lag terms of input digitization into the regression model to investigate the dynamic effect of digital input on carbon emission reduction. Columns (1), (2), (3), and (4) of Table 7 show the impact of the current term and the first, second, and third lag term of manufacturing’s input digitization on carbon emission intensity. With the increase in the lag period, digital inputs’ effects on carbon emission reduction increase. The main reason is that the manufacturing industry’s digital transformation is a long-term process. The increase in digital input will have a certain effect in improving production efficiency and energy efficiency in the short term. However, the technology spillover effect caused by information sharing and the high-level heterogeneous production mode requires continuous R&D investment and technology accumulation. Therefore, as time goes on, the effect of carbon emission reduction in input digitization on manufacturing will continue to rise.

#### 5.5.2. The Spillover Analysis

Digital technology has a wide range of penetrating and influencing capabilities, which can achieve spillover effects in other manufacturing industries through industrial correlation and the penetration effect [44] and then exert influence on the carbon emissions of other industries. Therefore, this paper refers to the work of Su et al. [74] and uses the spatial econometrics method to test the impact of manufacturing’s input digitization on carbon emissions in other manufacturing industries by participating in the Global Value Chain (GVC). In this part, the paper uses the Spatial Dubin Model (SDM) to adjust the benchmark regression model. The specific model is set as follows:(12)$$Carbon_{cit}=\beta DIG_{cit}+\theta WDIG_{cit}+\gamma Controls_{cit}+u_{c}+u_{i}+u_{t}+\varepsilon_{cit}$$
where *W* is the spatial weight matrix, and other variables share the same meaning as Equation (1). The design method for the spatial weight matrix also references Su et al. [74] to construct the forward and backward overflow matrix. The forward overflow matrixreflects the forward flow of value-added in GVC, which is set as:(13)$$\begin{array}{l} W_{F}=(\hat{V}B\hat{Y}{)}^{T}- [\begin{array}{cccc} V^{1}B^{11}Y^{1} & 0 & \cdots & 0\\ 0 & V^{2}B^{22}Y^{2} & \cdots & 0\\ \vdots & \vdots & \ddots & \vdots \\ 0 & 0 & \cdots & V^{2}B^{22}Y^{2} \end{array}]\\ ={ [\begin{array}{cccc} V^{1}B^{11}Y^{1} & V^{1}B^{12}Y^{2} & \cdots & V^{1}B^{1G}Y^{G}\\ V^{2}B^{21}Y^{1} & V^{2}B^{22}Y^{2} & \cdots & V^{2}B^{2G}Y^{G}\\ \vdots & \vdots & \ddots & \vdots \\ V^{G}B^{G1}Y^{1} & V^{G}B^{G2}Y^{2} & \cdots & V^{G}B^{GG}Y^{G} \end{array}]}^{T}- [\begin{array}{cccc} V^{1}B^{11}Y^{1} & 0 & \cdots & 0\\ 0 & V^{2}B^{22}Y^{2} & \cdots & 0\\ \vdots & \vdots & \ddots & \vdots \\ 0 & 0 & \cdots & V^{2}B^{22}Y^{2} \end{array}]\\ = [\begin{array}{cccc} 0 & V^{2}B^{21}Y^{1} & \cdots & V^{G}B^{G1}Y^{1}\\ V^{1}B^{12}Y^{2} & 0 & \cdots & V^{G}B^{G2}Y^{2}\\ \vdots & \vdots & \ddots & \vdots \\ V^{1}B^{1G}Y^{G} & V^{2}B^{2G}Y^{G} & \cdots & 0 \end{array}] \end{array}$$

Among them, $\hat{V}$ represents the diagonalization matrix of the value-added coefficient vector, *V*. *B* and $\hat{Y}$ have the same meaning as above, representing the Leontief inverse matrix and diagonalization matrix of the final product vector, *Y*, respectively. From the perspective of value suppliers, the forward overflow matrix, $W_{F}$, traces the forward connection between a manufacturing industry in one country and all downstream manufacturing industries in all countries, which can reflect the impact of input digitization in a manufacturing industry on the carbon emissions of downstream manufacturing industries.

Similarly, the backward overflow matrix, $W_{B}$, reflects the backward flow of value-added in GVC, which is set as:(14)$$\begin{array}{l} W_{B}=\hat{V}B\hat{Y}- [\begin{array}{cccc} V^{1}B^{11}Y^{1} & 0 & \cdots & 0\\ 0 & V^{2}B^{22}Y^{2} & \cdots & 0\\ \vdots & \vdots & \ddots & \vdots \\ 0 & 0 & \cdots & V^{2}B^{22}Y^{2} \end{array}]\\ = [\begin{array}{cccc} 0 & V^{1}B^{12}Y^{2} & \cdots & V^{1}B^{1G}Y^{G}\\ V^{2}B^{21}Y^{1} & 0 & \cdots & V^{2}B^{2G}Y^{G}\\ \vdots & \vdots & \ddots & \vdots \\ V^{G}B^{G1}Y^{1} & V^{G}B^{G2}Y^{2} & \cdots & 0 \end{array}] \end{array}$$

From the perspective of value-consumers, the backward overflow matrix, $W_{B}$, traces the backward connection between a manufacturing industry in one country and all upstream manufacturing industries in all countries, which can reflect the impact of input digitization in a manufacturing industry on the carbon emissions of upstream industries.

Column (5) and column (6) of Table 7 show the regression results under $W_{F}$ and $W_{B}$, respectively. The regression results show that the coefficients of *W* × *DIG* are both significantly negative, which illustrates that the increase in input digitization in the manufacturing industry positively impacts carbon emission reduction in both downstream and upstream manufacturing industries via participation in GVC. From the perspective of forward spillover, the possible reason is that embedding digital elements improves the quality and value-added of products, promotes innovation in the upstream manufacturing industry, and then realizes the overflow of technology, information, and digital elements to the downstream manufacturing industry through the provision of intermediate products. Moreover, the improvement of the informatization level brought on by the growth of digital input effectively reduces the cost of obtaining technical information among industries, which is conducive to the diffusion of advanced production technology and environmental protection technology from top to bottom. From the perspective of backward spillover, digital input improves the energy-use efficiency of the downstream manufacturing industry and realizes the replacement of pollution factors. The improvement of production mode and efficiency make the manufacturing industry less dependent on the intermediate input from upstream heavy industries with high pollution, which forces upstream enterprises to carry out green transformation. Simultaneously, the improvement of the information network makes it more convenient for downstream industries to feedback market and demand information to the upstream, which is good for avoiding invalid inventory and resource waste and eventually helps reduce the carbon emission intensity [75].

## 6. Further Research for Developing Countries

### 6.1. The Impact of Digital Input from Different Sources on Carbon Emission Intensity in Developing Countries

With the deepening of GVC division system, developing countries have undertaken most of the world’s polluting production activities. IEA statistics show that, in 2019, developing countries occupied 40.6% of the world’s total economy, while the total carbon emissions of developing countries account for 58% of the world. Developing countries are facing severe pressure regarding carbon emission reduction. Based on the analysis and hypothesis above, we will further examine input digitization’s impact on developing countries’ carbon emissions.

As shown in Table 8, column (1) displays the regression result of digital input from all economies for the carbon emission intensity of the manufacturing industry in developing countries. The results show that the increase in the digital input of manufacturing will significantly reduce the carbon emission intensity. However, compared with the benchmark regression, when all economies are selected as samples in column (8) of Table 3, the absolute values of the coefficient and significance levels both decreased. The reason may be that developing countries will spend more on R&D investment, equipment expenditure, and other early costs in digital transformation. The increase in cost will restrain the increase in enterprises’ environmental protection expenditures and limit digitization’s effect on carbon reduction. Furthermore, developing countries’ relatively insufficient human capital and weak environmental regulation shall also inhibit the technology spillover effect of digital input from other countries, resulting in a less obvious carbon emission reduction effect.

Column (2) of Table 8 shows the impact of digital input from developed countries on the carbon emission intensity of manufacturing in developing countries. After adding the square of digital input from developed countries, the coefficient of digital input from developed countries is significantly positive, while the coefficient of the quadratic term is significantly negative, which means that the digital input from developed countries and the carbon emission intensity of developing countries show an inverted U-shaped relationship. In other words, the digital input from developed countries will inhibit carbon emission reduction in developing countries at first, but when it reaches the threshold, the driving effect of digital input on carbon emission reduction will exceed the inhibitory effect and play a leading role, making it continue to promote a reduction in carbon emission intensity. Therefore, hypothesis 2 and hypothesis 3 are verified. In addition, column (3) displays that the impact of digital input from developing countries on the carbon emission intensity of the manufacturing industry in developing countries is not significant. On the one hand, the influence may be inhibited by the cost of digital transformation and the limited human capital in developing countries. On the other hand, the relatively low technical content of digital input from developing countries should also be considered, making the technology spillover effect weaker.

### 6.2. The Impact of Digital Input from Developed Countries on Carbon Emission Intensity in Developing Countries—Heterogeneity Analysis

In the last part of the paper, we conclude that the digital input from developed economies will lead to an upward and then downward trend in carbon emissions in developing countries. However, considering the differences in the types of manufacturing industries and the economic structures of developing countries, the effects of digital input from developed economies on carbon emissions in developing countries will also differ, so this paper will further perform an extended analysis.

#### 6.2.1. The Impact of Digital Input from Developed Economies on Carbon Emissions in Developing Countries Based on Different Types of Manufacturing Industries

Considering that the industrial transfer from developed economies has obvious industrial characteristics, this paper will specify the analysis of the digital input’s impact from developed economies on the carbon emissions of different manufacturing industries in developing countries. Columns (1), (2), and (3) of Table 9 show the effects of digital input from developed economies on the carbon emission intensity of labor-intensive, resource-intensive, and capital-intensive industries in developing countries, respectively. The empirical results show that digital input from developed countries leads to an upward and then downward trend in carbon emissions in either type of manufacturing industry, the logic of which is consistent with the previous analysis. This inverse U-shaped relationship is more obvious in the resource-intensive industries, indicating that the polluting production activities transferred from developed countries are most prominent in resource-intensive industries, leading to negative environmental effects in the short term. At the same time, with the improvement of production efficiency brought on by digital input and weak environmental regulations, the scale of polluting industries has expanded rapidly. In the long term, with the absorption and utilization of digital technology, the positive environmental effect of digital technology gradually becomes prominent, and digital elements gradually achieve substitution for polluting elements, eventually realizing beneficial environmental effects.

#### 6.2.2. The Impact of Digital Input from Developed Economies on Carbon Emissions in Developing Countries Based on Different Economic Structures

Considering the different economic development models and the ways of taking over manufacturing industries from developing countries, we will further distinguish two groups of developing economies according to different economic industrial structures and test the effects of digital input from developed economies on their carbon emissions. The first group is the typical industrialized countries, including China, Brazil, India, Russia, and Mexico, and the second group is the non-traditional industrialized countries, including Bulgaria, Indonesia, Romania, and Turkey. Columns (4) and (6) in Table 9 display the impact of digital inputs from developed countries on carbon emissions in these two groups of economies, respectively, and columns (5) and (7) add the square term of digital inputs from developed countries to them, respectively. The results show that, for the first group of developing countries, the impact of digital input from developed countries on their carbon emissions still shows an upward and then downward trend. The industrialization paths of China, Brazil, India, and Mexico are all promoted by taking over a large number of manufacturing industries from developed countries due to their abundant labor, land, energy, and other resource advantages. Although Russia has a good industrial foundation, its abundant minerals and energy will also lead to the migration of polluting manufacturing industries, so the inverted U-shaped characteristics of the impact of digital inputs on carbon emissions are particularly significant in these countries. Columns (3) and (4) show that, for non-traditional industrialized countries, digital input from developed countries significantly reduces manufacturing emissions because manufacturing is not the mainstay of these countries, while their agriculture and tertiary industries, such as tourism, are relatively advanced and do not possess the geographic conditions to attract polluting manufacturing industries to move in. Therefore, the suppressive effect on the environment of developed economies’ digital inputs is relatively small.

## 7. Conclusions and Further Discussions

### 7.1. Conclusions

Based on the panel data of 40 major economies in the world from 2005–2011, this paper uses a fixed-effect model to empirically test the impact of the input digitization of manufacturing on the complete carbon emission intensity and conduct extended analyses from multiple angles, and it also examines the impact of digital input on the carbon emissions of developing countries based on the differences between input sources. The main conclusions are as follows:(1)The increase in input digitization degree in the manufacturing industry significantly reduces the complete carbon emission intensity. The effect of carbon emission reduction gradually enhances as time goes on, with an obvious industry spillover effect. A mechanism test shows that it is mainly realized through the improvement of production efficiency, energy efficiency, and information transmission. The conclusion is still robust after replacing the core explanatory variables or the explained variables, considering the endogenous problems, and changing the sample period.(2)The effect of input digitization on carbon emission reduction is the most obvious when selecting resource-intensive industries as regression samples, followed by labor-intensive industries, while the environmental benefits of digital input on capital-intensive industries are not significant, which indicates that the effect of carbon emission reduction on digital input is more obvious to industries with lower technical levels and more pollution factors. In addition, input digitization’s effect on the manufacturing industry’s carbon emission reduction is directly proportional to the digital level of input economies.(3)When taking developing countries as research subjects, it is found that the carbon reduction effect of digital input from all economies is relatively small and that digital input from developed countries has an inverted U-shaped relationship with the complete carbon emission intensity of developing countries. That is to say, in the short term, digital input from developed countries will increase the carbon emission intensity of the manufacturing industry in developing countries. After reaching a certain threshold, the inhibitory effect of digital input on carbon emission will be dominant, reducing the carbon emission intensity of developing countries. This finding is particularly significant when selecting industries with more polluting elements and traditional industrialized countries as regression samples. However, the impact of digital input from developing countries on the carbon emission intensity of the manufacturing industry in developing countries is not obvious.

### 7.2. Suggestions

Against the background of the third scientific and technological revolution, the rapid development of digital technology provides a new idea for accelerating global carbon emission reduction. Based on the conclusions above, this paper puts forward the following countermeasures and suggestions:(1)The manufacturing industry should accelerate digital transformation and promote the deep integration of digital elements. The industry is supposed to introduce advanced manufacturing digital equipment and use new digital technologies to realize the improvement of production efficiency and structure rationalization. For example, the manufacturing industry could build a digital supply chain system and gradually turn to a consumer-led manufacturing mode. Furthermore, enterprises should fully use new digital technologies’ learning and absorption ability, accelerating the research and absorption of advanced production or environmental protection technologies to improve energy efficiency. Furthermore, the government is required to strengthen the construction of digital infrastructure and speed up the establishment of data aggregation networks, promoting information system integration and data resource sharing, so as to make full use of the technology spillover effect of the digital network and eventually achieve pollution control and carbon neutralization.(2)In order to maximize the effect of carbon emission reduction brought on by input digitization, we should focus on the digital transformation of the manufacturing industry with intensive pollution factors and accelerate the digital and intelligent transformation of production lines, workshops, and factories to optimize production and operation structures. The industry should strengthen the control of energy utilization, use digital methods to monitor and collect information on energy consumption, accurately customize energy use plans after network sharing and data analysis, and maximize the substitution of digital elements for pollution elements.(3)From the perspective of developing countries, the government should strictly control the pollution emissions of the transferred industry from developed countries and make efficient use of the accompanying digital inputs. On the one hand, a strict screening method is required. Developing countries are bound to monitor the pollution situation of transferred industries and strengthen the cooperation of departments and public supervision to avoid development at the expense of the environment and minimize the adverse environmental effects of digital input from developed countries in the short term. On the other hand, the government should encourage the introduction of knowledge-intensive industries with high-tech digital input and fully utilize its advanced technology. The government should increase the support for education expenditure and attach importance to the cultivation of human capital to strengthen the absorption of advanced digital technology spillover from developed countries. Moreover, developing countries should establish their emerging digital industries and realize initiating innovation through imitation, and, finally, foster local advantageous digital industries and achieve carbon emission reduction in the long term.

### 7.3. Limitations and Further Research

This study also has certain limitations and requires further research. Firstly, due to the limitation of data acquisition, the sample period of this paper is selected relatively early. Although the data are still powerful in explaining the impact of input digitization on carbon emission, if the data could be updated, their explanatory power will be stronger. Secondly, it is difficult for the paper to provide more detailed intermediary mechanism tests because of the restricted data acquisition, such as promoting technological innovation and technological spillover effects, though the theoretical analysis and mechanism tests in this paper can include these contents. Thirdly, due to space limitations, this paper only analyzes the heterogeneous effects of digital inputs from economies with different development levels and digital levels. In fact, digital inputs can be classified according to other criteria, such as the characteristics of digital industry and input purpose. Based on this study, follow-up research may further refine the research.

## Figures and Tables

**Figure 1 ijerph-19-12855-f001:**
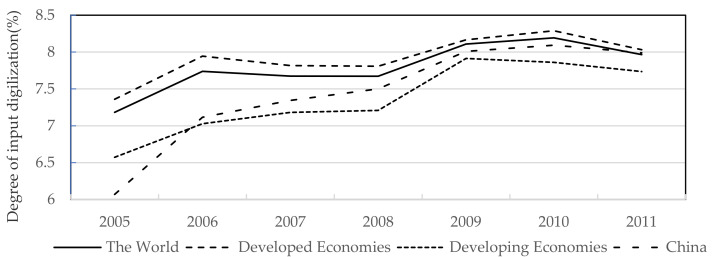
The trend of the manufacturing industry’s input digitization from different economies in the world.

**Figure 2 ijerph-19-12855-f002:**
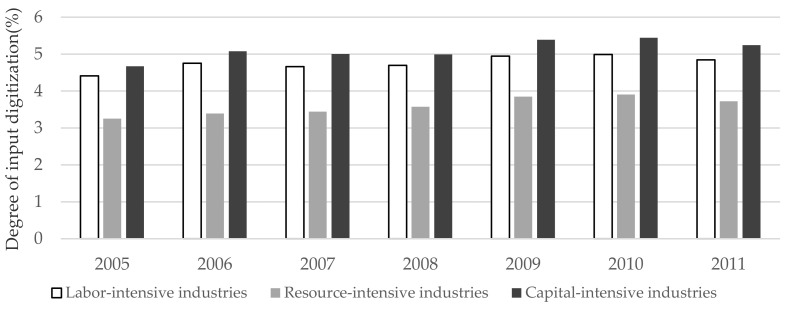
The trend of different types of manufacturing industries’ input digitization in the world.

**Figure 3 ijerph-19-12855-f003:**
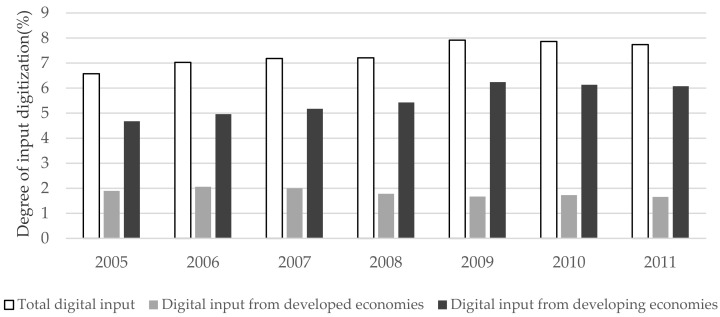
Distribution of different sources of input digitization in developing economies.

**Table 1 ijerph-19-12855-t001:** Division of digital industry.

Digital Industry	Industries (under ISIC Rev3.1 Classification Standard)	Split By
Computer and communication equipment manufacturing	30–33—Manufacturing of office, accounting, and computing machinery; manufacturing of radio, television, and communication equipment and devices; manufacturing of television, computer, radio transmitters, cable telephone, and telegraph equipment	The proportion of ICT product trade to total product trade
Computer software services	72—Computer and related activities: 7210 Hardware consulting; 7221 Software publishing; 7229 Consultation and supply of other software; 7230 Data processing; 7240 Database activities and online distribution of electronic content; 7290 Other computer-related activities	—
Electronic postal and telecommunication services	64—Post and telecommunications: 641—Postal activities; 642—Telecommunications: wired, wireless, satellite	The proportion of service trade in digital delivery mode to the total service trade
Internet publishing	22—Publishing activities: 2219—Other publications	The proportion of trade volume of digital publishing products to the trade volume of the sector
Online wholesale	51—Wholesale trade	The proportion of e-commerce industry scale to wholesale industry scale
Online retail	52—Retail trade	The proportion of e-commerce industry scale to retail industry scale

**Table 2 ijerph-19-12855-t002:** Classification of economies and manufacturing industries.

Category	Classification
Developed economies	Australia, Austria, Belgium, Canada, Cyprus, Czech Republic, Germany, Denmark, Spain, Estonia, Finland, France, United Kingdom, Greece, Hungary, Ireland, Italy, Japan, South Korea, Lithuania, Luxembourg, Latvia, Malta, Netherlands, Poland, Portugal, Slovakia, Slovenia, Sweden, Taiwan of China, USA
Developing economies	Bulgaria, Brazil, China, Indonesia, India, Mexico, Romania, Russia, Turkey
High-digital-level economies	Switzerland, Denmark, Luxembourg, Netherlands, Japan, Belgium, Canada, USA, Estonia, Spain, UK, France, Germany, Finland
Medium-digital-level economies	South Korea, Taiwan of China, Latvia, Slovakia, Ireland, Lithuania, Austria, Hungary, Slovenia, Czech Republic, Australia, Poland, Malta
Low-digital-level economies	Portugal, Russia, Cyprus, Romania, Greece, Italy, Bulgaria, Mexico, Brazil, Turkey, India, China, Indonesia
Manufacturing industries	C3: Food, Beverages, and Tobacco; C4: Textiles and Textile Products; C5: Leather, Leather, and Footwear; C6: Wood and Products of Wood and Cork; C7: Pulp, Paper, Paper, Printing, and Publishing; C8: Coke, Refined Petroleum, and Nuclear Fuel; C9: Chemicals and Chemical Products; C10: Rubber and Plastics; C11: Other Non-Metallic Minerals; C12: Basic Metals and Fabricated Metal; C13: Machinery, Nec; C14: Electrical and Optical Equipment; C15: Transport Equipment; C16: Manufacturing, Nec; Recycling
Labor-intensive manufacturing	C3, C4, C5, C6
Resource-intensive manufacturing	C8, C9, C10, C11, C12
Capital-intensive manufacturing	C7, C13, C14, C15, C16

**Table 3 ijerph-19-12855-t003:** Benchmark regression results.

Variables	(1)	(2)	(3)	(4)	(5)	(6)	(7)	(8)
DIG	−0.166 **	−0.065 **	−0.395 ***	−0.380 ***	−0.356 ***	−0.351 ***	−0.329 ***	−0.283 ***
	(−1.49)	(−0.58)	(−3.94)	(−3.79)	(−3.61)	(−3.56)	(−3.32)	(−2.89)
Va		−0.050 ***	−0.079 ***	−0.085 ***	−0.071 ***	−0.072 ***	−0.072 ***	−0.064 ***
		(−9.57)	(−12.90)	(−13.27)	(−10.98)	(−11.20)	(−11.21)	(−9.98)
Dvafs			0.028 **	0.031 **	0.031 **	0.030 **	0.032 **	0.024 *
			(2.18)	(2.40)	(2.44)	(2.37)	(2.53)	(1.91)
CL				−0.021 ***	−0.021 ***	−0.022 ***	−0.023 ***	−0.020 ***
				(−3.28)	(−3.23)	(−3.41)	(−3.54)	(−3.16)
Energy					0.049 ***	0.046 ***	0.046 ***	0.043 ***
					(10.39)	(9.33)	(9.36)	(8.96)
Construct						0.067 ***	0.060 ***	0.066 ***
						(3.34)	(2.92)	(3.27)
FDI							−0.011 **	−0.017 ***
							(−2.21)	(−3.27)
Env								0.207 ***
								(8.85)
Constant	0.598 ***	0.929 ***	1.296 ***	1.542 ***	1.371 ***	1.367 ***	1.380 ***	0.767 ***
	(58.16)	(17.66)	(20.03)	(17.74)	(15.37)	(15.34)	(15.53)	(8.67)
Country fixed effect	yes	yes	yes	yes	yes	yes	yes	yes
Industry fixed effect	yes	yes	yes	yes	yes	yes	yes	yes
Time fixed effect	yes	yes	yes	yes	yes	yes	yes	yes
Observations	3920	3920	3920	3920	3920	3920	3920	3920
$R^{2}$	0.391	0.407	0.485	0.486	0.502	0.504	0.504	0.515

Note: standard errors are in parentheses; *, **, and *** represent the 10%, 5%, and 1% significance levels, respectively.

**Table 4 ijerph-19-12855-t004:** Endogenous treatment and robustness test.

Variables	(1)	(2)	(3)	(4)	(5)
	Carbon	${\mathrm{Carbon}}_{d}$	Carbon	Carbon	Carbon
DR	−0.968 ***				
	(−5.41)				
DIG		−0.059 **	−2.177 ***	−2.822 ***	−0.133 **
		(−2.19)	(−5.14)	(−5.21)	(−1.97)
Unidentifiable test			35.727 ***	24.128 ***	
Weak instrumental variable test			865.452 ***	30.044 ***	
Observations	3920	3920	3920	3920	2800
$R^{2}$	0.520	0.154	0.290	0.291	0.119

Note: ** and *** represent the 5% and 1% significance levels, respectively.

**Table 5 ijerph-19-12855-t005:** Mechanism test.

Variables	(1)	(2)	(3)	(4)	(5)	(6)	(7)
	Carbon	OL	Carbon	Energy	Carbon	Inf	Carbon
DIG	−0.283 ***	2.350 ***	−0.250 **	−0.646 *	−0.283 ***	3.200 ***	−0.451 ***
	(−2.89)	(12.67)	(−2.51)	(−1.76)	(−2.89)	(5.39)	(−4.00)
OL			−0.017 ***				
			(−2.49)				
Energy					0.043 ***		
					(8.96)		
Inf							−0.036 ***
							(−15.11)
Constant	0.767 ***	−0.347 ***	0.689 ***	1.523 ***	0.767 ***	5.016 ***	1.000 ***
	(8.67)	(−2.07)	(8.62)	(54.15)	(8.67)	(9.36)	(9.84)
Control variables	yes	yes	yes	yes	yes	yes	yes
$R^{2}$	0.515	0.544	0.515	0.147	0.515	0.644	0.341
Sobel Z		−11.67 ***		−2.847 ***		−14.53 ***	
95% confidence interval		−0.806~−0.502		0.027~0.259		−1.262~−0.854	
Observations	3920	3920	3920	3920	3920	3920	3920

Note: *, **, and *** represent the 10%, 5%, and 1% significance levels, respectively.

**Table 6 ijerph-19-12855-t006:** Heterogeneity test.

Variables	(1)	(2)	(3)	(4)	(5)	(6)
	Labor	Resource	Capital	Carbon	Carbon	Carbon
DIG	−0.354 ***	−2.810 ***	−0.066			
	(−3.00)	(−4.14)	(−0.65)			
${\mathrm{DIG}}_{H}$				−0.695 ***		
				(−2.81)		
${\mathrm{DIG}}_{M}$					−0.663 ***	
					(−2.96)	
${\mathrm{DIG}}_{L}$						−0.212 *
						(−1.85)
Control variables	yes	yes	yes	yes	yes	yes
Observations	1120	1400	1400	3920	3920	3920
$R^{2}$	0.561	0.368	0.633	0.315	0.526	0.525

Note: * and *** represent the 10% and 1% significance levels, respectively.

**Table 7 ijerph-19-12855-t007:** Dynamic analysis and spillover analysis.

Variables	(1)	(2)	(3)	(4)	(5)	(6)
	Carbon	Carbon	Carbon	Carbon	Carbon	Carbon
DIG	−0.283 ***				−0.495 *	−0.408 *
	(−2.89)				(−1.87)	(−1.67)
${\mathrm{DIG}}_{1}$		−0.280 ***				
		(−2.92)				
${\mathrm{DIG}}_{2}$			−0.368 ***			
			(−3.67)			
${\mathrm{DIG}}_{3}$				−0.549 ***		
				(−4.85)		
*W*×*DIG*					−1.441 *	−1.089 **
					(−1.73)	(−2.34)
Control variables	yes	yes	yes	yes	yes	yes
Observations	3920	3360	2800	2240	3920	3920
$R^{2}$	0.520	0.154	0.290	0.119	0.568	0.482

Note: *, **, and *** represent the 10%, 5%, and 1% significance levels, respectively.

**Table 8 ijerph-19-12855-t008:** Regression results for developing countries.

Variables	(1)	(2)	(3)
DIG	−0.100 *		
	(−0.59)		
${\mathrm{DIG}}_{\mathrm{FD}}$		0.269 ***	
		(2.77)	
${{\mathrm{DIG}}_{\mathrm{FD}}}^{2}$		−1.104 **	
		(−2.47)	
${\mathrm{DIG}}_{\mathrm{FZ}}$			−0.463
			(−0.68)
Control variables	yes	yes	yes
Observations	882	882	882
$R^{2}$	0.712	0.711	0.433

Note: *, **, and *** represent the 10%, 5%, and 1% significance levels, respectively.

**Table 9 ijerph-19-12855-t009:** Regression results for developing countries—heterogeneity analysis.

Variables	(1)	(2)	(3)	(4)	(5)	(6)	(7)
	Labor	Resource	Capital	Carbon	Carbon	Carbon	Carbon
${\mathrm{DIG}}_{\mathrm{FD}}$	0.271 **	1.704 **	0.441 ***	0.117	0.795 ***	−0.267 ***	−0.239 *
	(2.27)	(2.3)	(3.18)	(1.31)	(3.86)	(−3.54)	(−1.87)
${{\mathrm{DIG}}_{\mathrm{FD}}}^{2}$	−0.826 *	−38.462 *	−2.570 ***		−3.904 ***		0.881
	(−1.68)	(−1.67)	(−3.51)		(−3.64)		(1.50)
Control variables	yes	yes	yes	yes	yes	yes	yes
Observations	315	315	392	490	490	392	392
$R^{2}$	0.653	0.517	0.480	0.823	0.828	0.758	0.794

Note: *, **, and *** represent the 10%, 5%, and 1% significance levels, respectively.

## Data Availability

Publicly available datasets were analyzed in this study. The data can be found here: https://www.rug.nl/ggdc/valuechain/wiod/wiod-2013-release; https://unctadstat.unctad.org/EN/ReleaseCalendar.html; https://unctadstat.unctad.org/EN/ReleaseCalendar.html; https://v2.fangcloud.com/share/a26979974d538c7e5aeb24b55a?folder_id=63000172546&lang=en, accessed on 9 August 2022.

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
