# Peer review of "Input Digitization of the Manufacturing Industry and Carbon Emission Intensity Based on Testing the World and Developing Countries"

_ijerph, 2022, doi:10.3390/ijerph191912855_

Round 1
Reviewer 1 Report
1. The conclusion and recommendation sections can be combined to further optimize the structure of the paper.
2. The introduction of the paper could be further developed and currently lacks an introduction to the overall framework content of the article.
3. The abstract is not written in a standardized manner and does not clearly identify the context and value of the paper, nor does it clearly state the empirical design of the paper and the research methods used.
4. This article lacks a section on research gaps and future perspectives and needs to be supplemented.
5. Table 7 and Table 8 in this paper are not specifically named in the header and need further clarification.
6. The format of the paper needs to be further standardized, e.g. formula fonts, lack of clarity in Figure 1, Figure 2, and Figure 3.
Author Response
Much obliged for your precious comments. In response to the points put forward by the reviewers, we put our replies in the following documents, please see the attachment.

Reviewer 2 Report
The paper examines an up-to-date research question: How does input digitalization affect carbon emissions. The paper contains a literature review section in which some relevant studies are discussed. Empirical analysis covers a series of robustness tests.
I have the following comments on the paper:
(1) The underlying theories that predict the impact of input digitalization on energy intensity are not explicitly discussed. The paper discussed mechanisms that conduct the impact of input digitalization: productivity, energy efficiency, and lower cost of information transformation. But what are the underlying theories that justify these channels through which input digitalization impacts energy intensity?
(2) The sample period covered by the paper is 2005-2011. I was wondering why this sample period can address the research question given that we are already in 2022. The data based on which the paper conducts their empirical analysis is more than 10 years ago. Digitalization in manufacturing firms is now very different from the situation 10 years ago. Digitalization, big data, and AI have been applied in manufacturing activities extensively in the past decade. The paper, although addresses an interesting question, does not really cover the period in which manufacturing digitalization really changes energy intensity.
(3) The three hypotheses set up by the paper also need some further considerations. In the introduction section and the hypothesis section, the paper seems to emphasize how FDI (in fact it should be FDI in manufacturing digitalization) affects energy intensity. The paper treats the test on the FDI impact the same as the benchmark test (impact of digitalization on energy intensity in general). Hence, the paper sets up three hypotheses parallelly. However, in their empirical analysis, the paper only spent a small paragraph on testing the FDI effect (H2&H3). The paper treats the FDI test equally with many other robustness tests, including the test on different manufacturing industries, the test on spillovers across industries, the test on countries with different level of digitalization, etc. Therefore, it is not clear what the main moderating factor is. Readers would ask the question why the paper did not set up hypotheses on the spillovers effect, on countries with different digitalisation levels, on different types of manufacturing industry? Why did the paper choose to set up hypothesis specifically for the FDI effect? If the hypotheses on the FDI effect are explicitly set up, the paper should treat FDI as the key moderating factor in changing the impact of digitalization on energy intensity, which should be reflected in the empirical analysis of the paper.
(4) The paper carried out a test on the mechanisms (Table 5 on page 15). In this test, all the results could be driven by multicollinearity between each pair of variables, and there are no reported results regarding other control variables in the regression. The endogenous problem exists in all regressions, especially in Table 5. There are other ways of testing the channels through which digitalization affects energy intensity, for example, by using interactive terms.
(5) In section 5.2.3. the paper explicitly deals with the endogeneity problem by using an IV technique. But the authors did not have a proper external IV, instead they use lagged terms of the endogenous variable. Hence, this is not a proper IV approach. In addition, there is no report of the results from the first stage IV estimation, which means that there is no way we know whether the IV used in the estimation is acceptable.
(6) In Table 8, the paper tests how digitalization in developed economies affects energy intensity in developing economies. The paper treats this test as the test for the FDI impact (Hypotheses 2&3). But one would ask how the aggregate digitalization of developed economies can reflect the FDI in manufacturing digitalization received by developing economies? To test the FDI impact, it requires the match of FDI in digitalization from developed economies with that received by recipient developing economies.
Overall, the paper examines an interesting research question. However, the paper lacks underlying theories, the sample period cannot reveal the recent practice of manufacturing digitalization, and there exist multiple issues in empirical analysis of the paper.
Author Response

(The authors gave the same response as above.)

Round 2
Reviewer 2 Report
The paper has improved in terms of the theoretical justification for the three channels through which digitalization in manufacturing reduces energy intensity. The paper also improved regarding the treatment of the endogeneity problem.
The paper can be further improved by considering the following comments:
(1) The paper should explicitly discuss what the paper means by digitalisation in manufacturing industry. In difference places the paper talks about big data, cloud computing, and AI, etc., but the paper’s empirical analysis does not cover these most recent digital technologies. The paper needs to make it clear what they investigate is ICT, otherwise readers will be confused about what the paper is about.
(2) Given that the paper cannot use up-to-date data on digitalisation, they should also make it explicit about (a) why the data they used is before 2011, (b) why it is still useful to examine the date 10 years earlier, and (c) what the implication can be drawn from using this data on ICT to the application of the most updated technologies in manufacturing sector, such as AI, cloud computing, etc.
(3) The paper can be more concise in many places. I find that some paragraphs are difficult to read and it is not easy to understand what the paper wants to say.
